# Obesity-Related Kidney Disease: A Growing Threat to Renal Health

**DOI:** 10.3390/ijms26146641

**Published:** 2025-07-10

**Authors:** Juan León-Román, Marina López-Martínez, Alexandra Esteves, Andreea Ciudin, Sara Núñez-Delgado, Tiffany Álvarez, Albert Lecube, Jorge Rico-Fontalvo, María José Soler

**Affiliations:** 1Nephrology Department, Vall d’Hebron University Hospital, Vall d’Hebron Institute of Research, CSUR National Unit of Expertise for Complex Glomerular Diseases of Spain, Paseo de la Vall d´Hebron 119, 08035 Barcelona, Spain; marina.lmga@gmail.com (M.L.-M.);; 2Department of Medicine, Universitat Autònoma de Barcelona, 08193 Bellaterra, Spain; 3Nephrology Department, Hospitais da Univerisdade de Coimbra, Unidade Local de Saúde de Coimbra, 3004-561 Coimbra, Portugal; 4Endocrinology Department, Vall d’Hebron University Hospital, Vall d’Hebron Institute of Research, Diabetes and Metabolism Research Group, Paseo de la Vall d´Hebron 119, 08035 Barcelona, Spain; 5CIBERDEM (Instituto de Salud Carlos III), 28029 Madrid, Spain; 6Asociación Colombiana de Nefrología e HTA, Bogotá 0814, Colombia; 7Facultad de Medicina, Departamento de Nefrología, Universidad Simón Bolívar, Carrera 59 Número 59-65, Barranquilla 111321, Colombia

**Keywords:** kidney disease, obesity, cardiovascular disease

## Abstract

Obesity represents a serious and growing disease worldwide. The pathophysiological changes secondary to chronic inflammation lead to the development of diseases that increase the morbidity and mortality of individuals. Chronic kidney disease (CKD) is a condition with deleterious effects that acts bidirectionally with obesity. From approximately 20% to 30% of individuals share phenotypes of CKD and obesity, increasing their cardiovascular risk and the risk of other complications. Obesity and CKD form a vicious cycle in which inflammation is the central axis of multiorgan damage. Despite increasing the risk of cardiac and renal mortality, CKD progresses in relation to body mass index and albuminuria. Nowadays, the implementation of the new medications aimed at mitigating the peak of inflammation is becoming a cornerstone of treatments for obesity, diabetes, cardiovascular diseases, and renal disease.

## 1. Introduction

Obesity is a complex and chronic disease that can potentially affect the quality of life by decreasing physical health as well as mental and social well-being. Despite the preventive interventions developed in recent years, the global prevalence and incidence of obesity continue to rise, to the extent that the World Health Organization (WHO) considers it a global health epidemic [1]. In 2022, approximately 2.5 billion people were affected by overweightness, including 890 million individuals with obesity. Currently, one in eight people has obesity, indicating that obesity rates double among adults and quadruple among adolescents compared to values from the 1990s [2]. In some areas, the prevalence of obesity is expected to double by 2050 [3].

Obesity is characterized by the excessive accumulation of adipose tissue, which may be attributed to genetic and/or socio-environmental factors, such as genetic predispositions, a sedentary lifestyle, high-calorie foods, insufficient access to healthy foods, low educational attainment, and psychiatric disorders. [4] Classically, obesity is described by a body mass index (BMI) ≥ 30 kg/m^2^, in contrast to overweight, which is defined by a BMI between 25 and 30 kg/m^2^ [5]. Currently, the diagnosis of obesity remains controversial and is also a matter of significant interest for multidisciplinary medical discussion, as the use of BMI as the sole diagnostic marker promotes erroneous stratification in many patients, leading to overdiagnosis with potential negative socioeconomic effects. BMI does not provide information about the individual health of patients. Furthermore, the definition of obesity established by BMI excludes the phenotypes that precede it, which also increase metabolic risk, and for which preventive and corrective treatment strategies should be established [6]. Therefore, BMI alone is often an insufficient biomarker, as it does not fully capture cardiometabolic risk. The combination of BMI ≥ 25 kg/m^2^ and a waist-to-height ratio (WtHr) of >0.5 helps optimize the stratification of obesity risk [7,8]. 

There is sufficient evidence highlighting the role of obesity in metabolic syndrome, cardiovascular diseases, endocrine disorders, mental health conditions, neoplastic diseases, and renal diseases [9,10,11]. Visceral obesity is also associated with increased inflammation and the risk of all-cause mortality [12]. Currently, a new classification of obesity is referred to as adiposity-based chronic disease (ABCD), which is based on its etiology, degree of adiposity, and associated health risks. Concurrently, the complications of obesity are influenced by two pathological processes: fat mass disease, which involves mechanical and physical forces, and sick fat disease, which pertains to endocrine and immune responses. The latter is associated with the activation of inflammatory responses that can lead to organ damage [13].

Chronic kidney disease (CKD) is a public health priority and a global concern, as it is projected to be among the top five causes of death by 2040. In many countries, the prevalence of CKD is underestimated due to insufficient screening measures to detect functional and structural renal abnormalities [14]. The progressive increase in the number of patients with CKD is also explained by the rise in cases of hypertension, metabolic syndrome (MS), and diabetes mellitus (DM) [15]. Thereby, Chang et al. describe that elevated BMI, waist circumference, and waist-to-height ratio are independent risk factors for a decline in glomerular filtration rate (GFR) and mortality in individuals with and without prior kidney disease [16].

The objective of this review is to focus on the importance of inflammation and the pathophysiological processes involved in the relationship between obesity and the development of kidney disease. Additionally, the various existing treatments and those currently under clinical trials are discussed.

## 2. Mechanisms Underlying Chronic Kidney Disease Development in Obesity

Obesity is an independent risk factor for the development of kidney disease, as it creates an environment of intraglomerular hypertension conditioned by many pathways. García-Carro et al. defined three main groups explaining the pathophysiological mechanisms of kidney disease in obesity: the hemodynamic, adipose tissue-related, and insulin resistance-hyperinsulinism pathways [17]. It is important that hemodynamic, adipose tissue-related, and insulin resistance pathways interact with each other and adapt to factors related to other comorbidities, such as age and sex (Figure 1).

### 2.1. The Hemodynamic Pathway

The hemodynamics of individuals with obesity are compromised in part secondary to the pathological hyperfiltration process associated with increased metabolic demands, the activity of the renin–angiotensin–aldosterone system (RAAS), the affinity of angiotensin II (Ang-II) receptors, fluid overload, and the positive feedback from the sympathetic nervous system [17].

The pathophysiological process includes vasodilation of the afferent arterioles and vasoconstriction of the efferent arterioles, associated with a reduction in tubuloglomerular feedback responsible for the vasoconstriction of the afferent arterioles [18]. The alteration of the renal sympathetic nervous system is also explained by the activation of chemoreceptors in the carotid bodies and, consequently, increased sympathetic activity [19,20]. Moreover, proximal tubular sodium reabsorption is increased, leading to diminished distal sodium delivery and stimulating the macula to increase renin secretion, thereby contributing to the perpetuation of the vicious cycle of hypertension and fluid overload [21].

### 2.2. Adipose Tissue-Related Pathway

Excessive accumulation of fat in patients with obesity enhances the endocrine and paracrine capabilities of adipocytes [22]. Visceral adipocytes also contain angiotensinogen, and their activity directly depends on the increase in BMI related to fat [23]. Adipocytes are responsible for the secretion of adipokines. Among the most important adipokines are leptin, adiponectin, tumor necrosis factor-α (TNF-α), resistin, interleukin-6 (IL-6), and plasminogen activator inhibitor 1 [24].

Adiponectin typically facilitates the oxidation of fatty acids and plays a crucial role in the regulation of glucose metabolism. This adipokine is primarily secreted by adipose tissue and is present in the bloodstream. Adiponectin levels are directly associated with renal function. Its physiological effects are mediated through receptors located in various tissues, including the arterial endothelium, smooth muscle cells of the kidney, and capillary endothelium [25]. In patients with obesity, there is an independent inverse association between albuminuria and adiponectin levels in nondiabetic individuals with overweight or obesity [26]. Thus, low levels of adiponectin also correlate with impaired fatty acid metabolism and insulin resistance. Studies in mice have shown that the deletion of adiponectin is associated with podocyte dysfunction, interstitial fibrosis, and albuminuria [27].

Unlike adiponectin, leptin levels are elevated in individuals with obesity and CKD, which represents a greater risk of hypertension, inflammation, and fibrosis [28]. Leptin enhances hypertension through the activation of the RAAS and increased sympathetic activity. Leptin also enhances fatty acid oxidation, the secretion of inflammatory cytokines such as monocyte chemoattractant protein-1 (MCP-1), and the formation of the NLRP3 inflammasome, exacerbating renal inflammation and fibrosis [29]. Studies have found that hyperleptinemia may contribute to the development of glomerulosclerosis and exert profibrotic effects on mesangial cells [28].

The activation of the NLRP3 inflammasome has been described in diabetes and obesity-related glomerulopathy (ORG), as hyperlipidemia and hyperglycemia activate the inflammasome through reactive oxygen species, mitochondrial damage, and stress [30,31]. At the renal level, the inflammasome acts in both the tubule-interstitium and glomeruli, promoting albuminuria through fibrosis and podocyte effacement [32].

Finally, the intestinal microbiota has also been identified as a key factor in the development of diseases such as obesity, T2DM, CKD, cardiovascular diseases, and certain types of cancer, due to its regulatory role in energy and immune balance [33]. Intestinal dysbiosis can be both caused and exacerbated by uremia, making CKD a contributing factor that is part of the vicious cycle of ongoing damage to the microbiota, alongside pro-inflammatory processes [34]. It has been observed that children with obesity present elevated levels of Bacteroides compared to the control group, and that quantitative and qualitative alterations of the intestinal microbiota are common in patients with CKD and end-stage renal disease (ESRD) [35,36].

### 2.3. Insulin Resistance–Hyperinsulinism Pathway

Insulin resistance is directly related to the visceral fat mass of individuals and the secretory activity of adipokines. The pro-inflammatory state of obesity inhibits insulin receptor substrate 1 (IRS-1) signaling pathways in adipose and muscle tissue, as well as limits the activity of peroxisome proliferator-activated receptor gamma (PPARγ), which is responsible for the processes of fat storage and lipid synthesis in adipose tissue [37].

Excessive insulin secretion interferes with podocyte and cytoskeleton activities [38]. Podocytes express glucose transporters (GLUT1, GLUT2, GLUT3, GLUT4, and GLUT8) and components of the insulin signaling cascade, such as IRS and the insulin receptor. Podocytes are capable of glucose uptake in response to insulin stimulation through GLUT4, as GLUT4 is the glucose transporter most sensitive to such stimuli [39,40]. Under normal conditions, insulin induces the rapid translocation of GLUT4 to the plasma membrane, promoting cytoskeletal remodeling and contraction, thereby facilitating the physiological response to glomerular pressure. Renal damage is manifested by the inability to phosphorylate AKT, which prevents the translocation of GLUT4 to the plasma membrane following insulin stimulation and, consequently, decreases glucose uptake. The deficiency of GLUT4 also suppresses the mammalian target of rapamycin (mTOR) pathway, resulting in a deficit in nutrient sensing [41]. Furthermore, insulin promotes tubulointerstitial fibrosis by enhancing the formation of type IV collagen and TGF-β in the renal tubules [38].

## 3. Effect of Obesity on Kidney 

Obesity is one of the modifiable risk factors for the development of CKD. There is evidence linking obesity to CKD. According to reports, 14 individuals per 1000 adults in the United States have obesity associated with CKD. Additionally, between 20% and 30% of individuals with obesity suffer from kidney disease, and more than 20% of adults with ESRD are diagnosed with morbid obesity [42]. Furthermore, obesity is a factor that complicates access to kidney transplantation in patients with ESRD, as the risk of complications increases compared to recipients with a normal BMI [43,44]. While obesity is described as a risk factor for the development of kidney disease, there are phenotypes of obesity that also represent a risk for renal damage. In the meta-analysis conducted by Valizadeh et al., it is noted that healthy patients with overweight and obesity have a higher risk of renal dysfunction, refuting the previously established notion of the benignity of these phenotypes [45].

Remarkably, the previously mentioned inflammatory mechanisms contribute to changes in renal structure, both due to the accumulation of ectopic fat and the increase in fat within the renal sinus [46]. Furthermore, obesity is also a risk factor for renal lithiasis and renal neoplasms. Nephrolithiasis is associated with low urinary pH, increased urinary oxalate, uric acid, and other electrolytes, while insulin resistance promotes the production of insulin-like growth factor 1, which may exert stimulating effects on the growth of various types of tumor cells [47].

ORG represents the structural manifestation of renal damage directly attributable to excess body weight; histologically, it is characterized by structural alterations in both the glomerulus and the renal interstitium [48,49]. Despite the fact that the prevalence of obesity is continuously increasing, only a proportion of individuals with obesity develop ORG. This suggests that predisposing factors, such as genetic susceptibility or low nephron mass, modulate the individual vulnerability of patients to chronic damage [50]. The albuminuria associated with ORG may present with objective clinical findings such as hypertension and/or edema, or it may be asymptomatic [51]. Since the presence of albuminuria indicates existing renal damage, recent experimental studies are advancing transcriptomic analysis to facilitate the prompt diagnosis of ORG through non-invasive biomarkers [52].

ORG is characterized by glomerulomegaly, focal and segmental glomerulosclerosis (FSGS), and tubulointerstitial fibrosis. Glomerulomegaly is defined as the diffuse enlargement of glomerular size and is interpreted as a compensatory mechanism in response to increased demand for renal function [23]. Morphologically, there is an enlargement of the glomeruli compared to the mean glomerular diameter of patients without obesity, adjusted for age and sex [49]. Additionally, it is accompanied by mesangial expansion and podocyte hypertrophy with foot process fusion [23]. Perihilar predominant FSGS is characterized by partial and heterogeneous podocyte effacement, with proteinuria generally in the subnephrotic range. The clinical manifestations of ORG are often variable [53]. Obesity also promotes the deposition of lipids in mesangial cells, podocytes, and renal tubules, which in turn enhances fibrosis and tubulointerstitial atrophy [46,54]. Finally, metabolic stress, combined with factors such as hypertension or dysfunction of the renin–angiotensin axis, promotes fibrosis through the activation of TGF-β and other profibrotic pathways [23].

## 4. Obesity and Cardiovascular–Kidney–Metabolic Syndrome

Cardiovascular disease is a significant cause of morbidity and mortality in individuals with obesity. The cardiometabolic risk depends on the distribution of fat, with visceral adipose tissue representing the highest associated risk [55]. Remote and local adipose tissue also exert pro-atherogenic and pro-inflammatory effects in certain organs [56]. Obesity accelerates the process of atherosclerosis through various mechanisms, including insulin resistance and inflammation. Thus, atherosclerosis promotes the development of cardiovascular diseases, such as ischemic heart disease, stroke, and death [10].

The development of atherosclerosis begins in childhood, with endothelial dysfunction and damage to the tunica being its initial steps. Consequently, the endothelium expresses adhesion molecules such as vascular cell adhesion molecule-1 and MCP-1, which are responsible for recruiting inflammatory cells such as monocytes and lymphocytes. The recruited monocytes mature into macrophages that take up cholesterol particles. The resulting inflammatory response leads to the secretion of interleukins, which promote the synthesis of extracellular matrix, consolidating and propagating the development of atheromatous plaques [57].

Obesity is one of the risk factors for cardiovascular disease, as the pro-inflammatory state exacerbates vascular damage and the recruitment of inflammatory cells. Obesity is also associated with metabolic distress. Several studies have shown that a high BMI and/or the accumulation of abdominal fat increase cholesterol deposits in the coronary arteries and raise the risk of other comorbidities such as heart failure, atrial fibrillation, sleep apnea, and stroke [58].

Obesity, cardiovascular disease, and renal diseases are currently encompassed within a syndrome known as cardiovascular–kidney–metabolic syndrome (CKM) (Figure 2). The main objective of describing this new syndrome is to focus on the pro-inflammatory state within a set of pathologies with similar pathophysiological phenomena, stratify risk, and optimize treatments and preventive measures [59].

## 5. Obesity, Diabetes, and Their Link with Kidney Disease

DM has been rising over the past decade, with an estimated 643 million people diagnosed in 2023 [60]. However, this does not account for the undiagnosed patients with type 2 diabetes mellitus (T2DM) who are unaware that they have this condition, in part ascribed to delays in diagnosis or lack of access to diagnostic tools [42]. T2DM is the most common type of diabetes worldwide, accounting for 90% of cases, and is the result of decreased pancreatic beta-cell function and increased insulin resistance. Although genetics plays a role, it is now established that unhealthy lifestyles and lower socioeconomic status are among the main factors that contribute to the development of prediabetes and diabetes mellitus [61].

DM has a significant impact on quality of life and increases cardiovascular and overall mortality. There is a higher risk of macrovascular complications such as myocardial infarction, stroke, and peripheral vascular disease, while microvascular complications include diabetic retinopathy, diabetic nephropathy, and diabetic neuropathy [61,62]. The increase in adipose tissue in obesity leads to the previously mentioned pro-inflammatory state, promoting insulin resistance. As a result, the pancreas adjusts to insulin resistance by initially increasing insulin secretion. However, the microenvironment created by obesity—characterized by hypoxia, mitochondrial dysfunction, and fibrosis—increases oxidative stress, resulting in beta-cell dysfunction and eventually leading to reduced beta-cell mass [61].

Therefore, obesity and diabetes have a bidirectional and intertwined relationship that is enhanced by kidney disease: obesity decreases the function of beta cells, thereby increasing insulin resistance, which, in turn, leads to hyperglycemia and, consequently, to T2DM. Conversely, patients with T2DM who have higher baseline insulin resistance can contribute to obesity due to elevated insulin levels and increased hepatic gluconeogenesis [61]. There is also a correlation between obesity and DM in the risk of developing kidney and cardiovascular diseases [63].

## 6. Challenges of New Management Options for Obesity and Kidney Disease

### 6.1. Lifestyle Interventions and Traditional Drugs

Glucose monitoring, a healthy lifestyle, and regular exercise should be the first options and the cornerstones of treatment. Many overweight individuals and patients with obesity might also reverse this condition or delay disease progression [64].

Traditionally, the use of angiotensin-converting enzyme inhibitors (ACE inhibitors) or angiotensin II receptor antagonists (ARBs) has served as a fundamental treatment to slow the deterioration of renal function by reducing the state of renal hyperfiltration, as demonstrated by the RENAAL (losartan) and IDNT (ibersartan) studies. However, the residual risk, despite standard treatments, continues to be a risk factor for major adverse events [65,66].

### 6.2. The Emerging Treatments of Cardiovascular–Kidney–Metabolic Syndrome: Incretin-Based Therapies and Gliflozins

Glucagon-like peptide-1 (GLP-1) is a gastrointestinal peptide secreted by the intestinal tract that enhances insulin release and decreases glucagon concentration under normal physiological conditions. Therefore, it represents a class of drugs based on the entero-insular axis, capable of modulating insulinotropic activity [67]. GLP-1 receptor agonists work by decreasing gastric emptying and increasing the sensation of fullness, thereby improving weight loss in addition to lifestyle changes [68].

The benefits of GLP-1 are numerous. Firstly, it has been described as playing an important role in controlling inflammation and reducing endothelial dysfunction. Additionally, it improves lipid metabolism and lowers blood pressure due to its natriuretic effect [69,70]. Its cardiac benefits in reducing major adverse cardiac events (MACEs) have also been reflected in several studies, including the LEADER trial (Liraglutide and Cardiovascular Outcomes in Type 2 Diabetes) [71], SUSTAIN-6 (Trial to Evaluate Cardiovascular and Other Long-term Outcomes With Semaglutide in Subjects With Type 2 Diabetes) [72], REWIND trial (Dulaglutide and Cardiovascular Outcomes in Type 2 Diabetes) [73], HARMONY Outcomes (Effects of Albiglutide on Major Cardiovascular Events In Patients With Type 2 Diabetes Mellitus) [74], SELECT trial (Semaglutide and Cardiovascular Outcomes in Obesity without Diabetes) [75], and SOUL trial (Oral Semaglutide and Cardiovascular Outcomes in High-Risk Type 2 Diabetes) [76].

Additionally, the renal benefits of GLP-1 have been described in patients with obesity, with or without T2DM, through studies such as the AMPLITUDE-O trial (Cardiovascular and Renal Outcomes with Efpeglenatide in Type 2 Diabetes), which demonstrated a reduction in albuminuria and less deterioration of renal function in the efpeglenatide group [77]. Additionally, the AWARD-7 trial (Dulaglutide versus insulin glargine in patients with type 2 diabetes and CKD) demonstrated a reduction in insulin use among patients treated with dulaglutide as well as a lower incidence of the combined endpoint of progression to ESKD or a reduction in glomerular filtration rate [78]. The LEADER and SUSTAIN-6 trials showed a reduction in MACE as well as a decrease in the progression of CKD due to a reduction in albuminuria [71,72]. The FLOW trial (Effect of semaglutide versus placebo on the progression of renal impairment in people with type 2 diabetes and chronic kidney disease) demonstrated that subcutaneous semaglutide was associated with a risk reduction in major adverse renal events (MARE) and death from cardiovascular causes in patients with T2DM and CKD [79]. The SMART (Semaglutide in patients with overweight or obesity and chronic disease without diabetes: a randomized double-blind placebo-controlled clinical) trial also established that semaglutide treatment for 24 weeks resulted in a clinically meaningful reduction in albuminuria in patients with overweight/obesity and non-diabetic CKD [80].

Sodium-glucose cotransporter-2 inhibitors (SGLT2i), also known as gliflozins or flozins, have proven to be a fundamental pillar in the treatment of CKM syndrome due to their inhibition of sodium and glucose reabsorption in the proximal convoluted tubule, promoting urinary glucose excretion and osmotic diuresis [81,82]. The success of this class of drugs is based on an insulin-independent mechanism that involves a competitive interaction between the SGLT2 protein and glucose binding in the renal tubules, followed by the transport of glucose across the basolateral membrane into the bloodstream [83].

There are sufficient clinical trials demonstrating the effectiveness of SGLT2i in the treatment of hyperglycemia, while also enhancing blood pressure control, promoting weight loss, and reducing the risk of developing MACE [84]. From a renal perspective, the CREDENCE (Canagliflozin and Renal Outcomes in Type 2 Diabetes with Established Nephropathy Clinical Evaluation) [85], the DAPA-CKD (Dapagliflozin and Prevention of Adverse Outcomes in Chronic Kidney Disease) [86], and the EMPA-KIDNEY (Empagliflozin in Patients with Chronic Kidney Disease) trials have shown a reduction in MARE, including decreases in albuminuria, mortality, and progression to CKD [87].

Although it is not from a group of drugs primarily used to treat obesity, finerenone is a highly selective non-steroidal mineralocorticoid receptor antagonist; its binding blocks the recruitment of transcriptional coactivators involved in the expression of pro-inflammatory and profibrotic factors. The FIDELIO-DKD (Effect of Finerenone on Chronic Kidney Disease Outcomes in Type 2 Diabetes) and the FIGARO-DKD (Cardiovascular Events with Finerenone in Kidney Disease and Type 2 Diabetes) studies demonstrated a reduction in MACE and MARE, along with a significant reduction in albuminuria levels compared to the control group [88,89].

### 6.3. Bariatric Surgery and Alternative Weight Loss Procedures

Bariatric surgery leads to favorable long-term outcomes, including reductions in body weight, decreased cardiovascular disease risk, improved glycemic control, and enhanced quality of life [90]. This surgery is offered to individuals with a BMI of ≥40 kg/m^2^ who struggle to lose weight despite lifestyle changes and exercise, or to those with a BMI of ≥35 Kg/m^2^ who have obesity-related comorbidities such as hypertension, T2DM or MS. Several studies have demonstrated improvements in estimated GFR and inflammatory biomarkers, as well as remission of albuminuria, following bariatric surgery [91,92,93,94]. Nevertheless, these invasive procedures are often associated with long-term side effects [91,95,96]. Other procedures, such as gastric emptying systems or intragastric balloons, have been proposed to achieve weight loss and, consequently, better glycemic control. However, studies regarding the long-term efficacy and safety of these devices are scarce [97].

### 6.4. Emerging Frontiers

Currently, there are no obesity biomarkers associated with the progression of CKD. Albuminuria remains one of the most reliable markers, but its presence indicates established structural damage. The literature regarding other biomarkers is relatively scarce. Nevertheless, microRNAs (such as microRNA-802, microRNA-155, microRNA-130b, and microRNA-21) are emerging as potential biomarkers for both obesity and the progression of CKD, as they can regulate pathways that culminate in inflammation and fibrosis in renal tissue [52].

### 6.5. Challenges

Obesity is a multifactorial and complex disease with numerous implications and consequences. Once kidney disease is established, reversing the inflammation and fibrosis caused by obesity is a matter of significant interest. Finerenone is one of the few drugs that allows for the reduction of renal inflammation and fibrosis. Studies are being conducted to verify its efficacy in non-diabetic CKD patients (FIND-CKD trial) [98].

### 6.6. Future Directions

Given the rapid increase in obesity prevalence, newer medications have been proposed, with recent studies highlighting the role of growth/differentiation factor 15 (GDF15). GDF15 belongs to the transforming growth factor-beta family, and when overexpressed, it leads to reductions in body weight and food intake in obese mice and monkeys. A dual analogue drug (GLP-1–GDF15) is being developed and has already demonstrated reductions in body weight, food intake, triglyceride levels, and glucose levels in obese mice and monkeys. This drug may constitute an important tool against obesity and kidney disease [5].

## 7. Conclusions

Obesity is associated with diseases that increase the morbidity and mortality of individuals. While obesity is related to the development of cardiovascular diseases and their associated complications, CKD is of vital importance, as it represents the continuation of the vicious cycle of inflammation, with bidirectional deleterious effects. Currently, obesity-focused treatments also benefit patients with diabetes, cardiovascular disease, and CKD. The key to these new treatments is breaking the toxic cycle of inflammation to reduce its adverse effects in both the short and long term. The treatment of CKD syndrome aims to improve the associated comorbidities of obesity through a holistic approach.

## Figures and Tables

**Figure 1 ijms-26-06641-f001:**
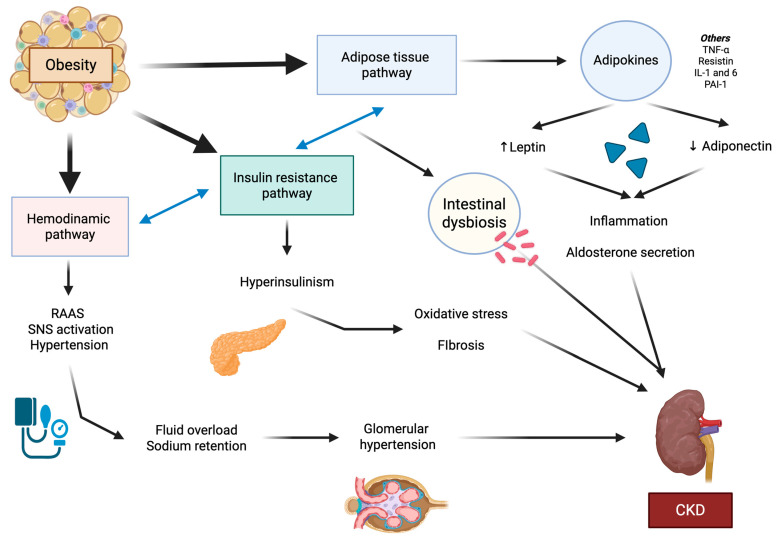
Pathophysiology of obesity. The mechanisms of renal disease in obesity. Three pathways have been described: hemodynamic, adipose tissue, and insulin resistance. These three pathways interact with each other, secreting adipokines and cytokines, activating the sympathetic nervous system, and promoting the pathological activation of the RAAS. All three pathways lead to renal damage, as the pro-inflammatory state and profibrotic factors favor glomerular hyperfiltration and, consequently, promote endothelial, podocyte, and tubular damage, increasing albuminuria excretion. RAAS: renin-angiotensin-aldosterone system. SNS: sympathetic nervous system. CKD: chronic kidney disease. TNF- α: tumor necrosis factor-α. IL: interleukin. PAI-1: plasminogen activator inhibitor 1.

**Figure 2 ijms-26-06641-f002:**
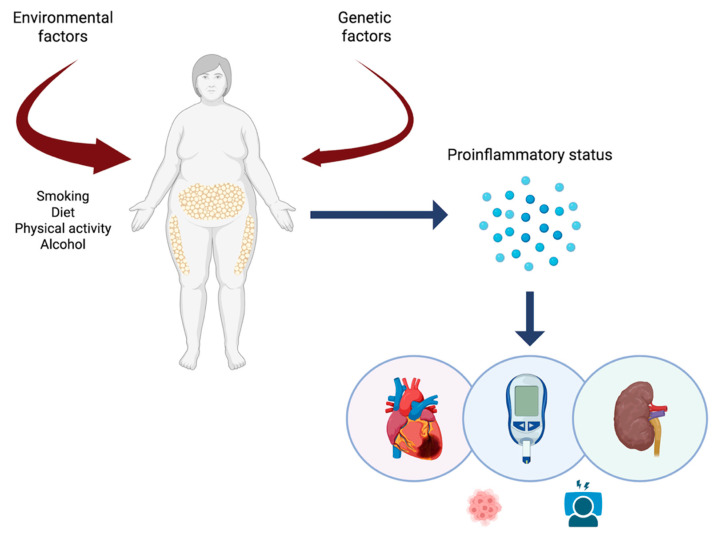
The cardiovascular–kidney–metabolic syndrome. The cardiovascular–kidney–metabolic (CKM) syndrome is the result of diseases affecting the organs and systems previously discussed, following exposure to environmental and/or genetic factors. These factors promote a pro-inflammatory state, triggering chronic pathologies based on inflammation and fibrosis. In addition to increasing cardiovascular and renal risk, obesity interferes with sleep physiology and increases the risk of developing neoplasms. All these clinical manifestations are part of a vicious cycle where inflammation is the cornerstone of the pathological process.

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
