# Peer review of "Obesity-Related Kidney Disease: A Growing Threat to Renal Health"

_ijms, 2025, doi:10.3390/ijms26146641_

Round 1

Reviewer 1 Report

Comments and Suggestions for Authors

The authors provide a concise and well-structured summary of the current understanding of obesity-related kidney diseases. While the manuscript is well written, there is considerable overlap with previously published reviews on this topic (https://pubmed.ncbi.nlm.nih.gov/37760939/).

I have outlined some suggestions below to strengthen the manuscript.

1) Obesity and Kidney subtopic title can be rephrased for clarity and thematic signposting

- subtopic appears to address structural and functional adaptations

2) A section on the following may be included

  1. a) emerging frontiers/research gaps
  2. b) future directions/perspectives
  3. c) challenges

this can include role of

  1. i) genetics/genomics/metabolomics
  2. ii) biomarkers

iii) phenotypes including potential role of pre-obesity since kidney disease may be initiated in the overweight phase. (https://pubmed.ncbi.nlm.nih.gov/39487860/)

The authors may also consider acknowledging the evolving shift in how clinical obesity is defined. There is a call to move beyond BMI (which may in fact have ethnic specific cut offs - e.g. in asians) as a sole metric for defining obesity. These should perhaps be factored in as a discussion point to support a more current discussion. (https://pubmed.ncbi.nlm.nih.gov/39824205/)

Author Response

We would like to express our gratitude to the reviewers and the editorial board for their efforts in reviewing our manuscript entitled “Obesity-Related Kidney Disease: A Growing Threat to Renal Health.” We appreciate the reviewers’ comments, which have undoubtedly enhanced the current version of the manuscript. Below is a detailed description of the changes made to the manuscript.

Reviewer: 1

1) Obesity and Kidney subtopic title can be rephrased for clarity and thematic signposting. Subtopic appears to address structural and functional adaptations.

We appreciate the reviewer's suggestion. We have replaced the initial title of “OBESITY AND THE DEVELOPMENT OF CHRONIC KIDNEY DISEASE” with “MECHANISMS UNDERLYING CHRONIC KIDNEY DISEASE DEVELOPMENT IN OBESITY”. We also rephrased the title of “OBESITY AND KIDNEY” to “EFFECT OF OBESITY ON KIDNEY DISEASE”.

2) A section on the following may be included: a) emerging frontiers/research gaps; b) future directions/perspectives; and c) challenges.

We appreciate the comment from the reviewer, we include the following paragraph at the bottom of the manuscript:

Emerging frontiers

Currently, there is no obesity biomarker that is associated with the progression of CKD. Albuminuria remains one of the most reliable markers, but its presence indicates established structural damage. The literature regarding other biomarkers is relatively scarce. Nevertheless, microRNAs (such as microRNA-802, microRNA-155, microRNA-130b, and microRNA-21) are emerging as potential biomarkers for both obesity and the progression of chronic kidney disease (CKD), as they can regulate pathways that culminate in inflammation and fibrosis in renal tissue (100).

Challenges

Obesity is a multifactorial and complex disease with numerous implications and consequences. Once kidney disease is established, the reversal of inflammation and fibrosis caused by obesity is matter of significant interest. Finerenone is one of the few drugs that allows for the reduction of renal inflammation and fibrosis. Studies are being conducted to verify its efficacy in non-diabetic CKD patients (FIND-CKD trial) (101).

Future directions

Given the rapid increase in obesity prevalence, newer medications have been proposed, with recent studies highlighting the role of growth/differentiation factor 15 (GDF15). GDF15 belongs to the transforming growth factor-beta (TGF-β) family, and when overexpressed, it leads to a reduction in body weight and food intake in obese mice and monkeys. A dual analogue drug (GLP-1–GDF15) is being developed and has already demonstrated reductions in body weight, food intake, triglycerides, and glucose levels in obese mice and monkeys. This drug may constitute an important tool against obesity and kidney disease (102).”

3) Phenotypes including potential role of pre-obesity since kidney disease may be initiated in the overweight phase.

Thank you very much for the comment made by the reviewer. We have added a paragraph in the section of EFFECT OF OBESITY ON KIDNEY DISEASE with the following text: While obesity is described as a risk factor for the development of kidney disease, there are phenotypes of obesity that also represent a risk for renal damage. In the meta-analysis conducted by Valizadeh et al., it is noted that healthy patients with overweight and obesity have a higher risk of renal dysfunction, refuting the previously established notion of the benignity of these phenotypes.

4) The authors may also consider acknowledging the evolving shift in how clinical obesity is defined. There is a call to move beyond BMI as a sole metric for defining obesity. These should perhaps be factored in as a discussion point to support a more current discussion.

We really appreciate the comment provided by the reviewer. We consider it as an important issue, so we have included the paragraph in the Introduction section with the following text: “Currently, the diagnosis of obesity remains controversial and is also a matter of significant interest for multidisciplinary medical discussion, as the use of BMI as the sole diagnostic marker promotes erroneous stratification in many patients, leading to overdiagnosis with potential negative socioeconomic effects. BMI does not provide information about the individual health of patients. Furthermore, the definition of obesity established by BMI excludes the phenotypes that precede it, which also increase metabolic risk, and for which preventive and corrective treatment strategies should be established”.

Reviewer 2 Report

Comments and Suggestions for Authors

The manuscript demonstrates that obesity and chronic kidney disease (CKD) are interrelated conditions that form a vicious cycle driven by chronic inflammation, significantly increasing cardiovascular risk and multiorgan damage. Consequently, emerging anti-inflammatory therapies are becoming key in managing obesity, diabetes, cardiovascular disease, and renal disease. However, there are several scientific concerns listed below that require careful verification. In addition, several minor errors also need to be addressed:

  1. The statement“Adiponectin is primarily found in...” appears to be a misunderstanding. Adiponectin is secreted by adipocytes and circulates in the bloodstream; it exerts its effects via receptors in tissues such as the arterial endothelium, kidney smooth muscle cells, and capillary endothelium.
  2. In the sentence“Insulin resistance is directly related to the fat mass of individuals and the secretory activity of adipokines”, does “fat mass” specifically refer to visceral fat mass? Please clarify.
  3. The sentence“…facilitating the filtration of albumin and promoting oxidative stress through NADPH oxidase and AKT/mTOR intracellular pathway or glucose transporter 4…” combines multiple mechanisms too tightly. Please briefly specify each mechanism. Additionally, please provide a reference supporting the direct involvement of GLUT4 in albuminuria or oxidative stress in the kidney.
  4. A reference is needed for the statement:“DM has been rising over the past decade, with an estimated 643 million people diagnosed in 2023.”
  5. For the statement “The pancreas adjusts to insulin resistance by initially increasing the number of beta-cells,” please provide a reference. It should be confirmed whether the typical compensatory mechanism involves increased insulin secretion rather than an increase in beta-cell number.
  6. In the sentence describing the FLOW trial,“The FLOW trial... demonstrated that subcutaneous semaglutide was associated with a risk reduction in MACE in patients with overweight or obesity and established cardiovascular disease without a history of diabetes”, please double-check the trial population. It appears this may be a confusion with the SELECT trial.
  7. “CKM syndrome”is a relatively new concept and is not yet a universally established clinical diagnosis. Please confirm whether SGLT2 inhibitors have been specifically indicated for CKM in the cited references.
  8. The mechanism of SGLT2 inhibitors does not involve Na⁺/K⁺-ATPase in glucose transport. Please verify and revise according to appropriate references.

  1. For the sentence “Obesity is a complex and chronic disease that can potentially affect the … and social well-being”, "their" doesn't have a clear plural referent since the subject is "obesity", which is singular.
  2. “Epidemic disease” is not the common phrasing used by WHO, please replace it.
  3. “Above 30 kg/m²” should be “≥30 kg/m²” for clinical accuracy.
  4. The tenses are mixed in the manuscript, for example, “was defined” and “is defined” is mixed in the sentence “obesity was defined by a body mass index … in contrast to overweight, which is 48 defined by… ”.
  5. “Neoplastic” should be part of a noun phrase in the sentence “There is enough evidence highlighting…”.
  6. “An increased inflammation” is incorrect → “increased inflammation” (uncountable).
  7. “Drop-in” should be “drop in” in the sentence “Thereby, Chang and cols. Described…”
  8. The sentence “Excessive insulin secretion favors insulin to interfere with the selective activity of podocytes...” is awkward
  9. pancreas typically compensates by increasinginsulin secretion, rather than the number of beta cells.
  10. For “The FLOW trial... demonstrated that subcutaneous semaglutide was associated with a risk reduction in MACE in patients with overweight or obesity and established cardiovascular disease without a history of diabetes”, please check patients involved in FLOW trial and whether you confused it with the SELECT trial.
  11. “CKM syndrome” is a relativelynewly defined construct, not a universally established clinical diagnosis. Has SGLT2i been specified for using in CKM in your reference?
  12. Themechanism of SGLT2 inhibitors is independent of Na⁺/K⁺-ATPase involvement in glucose transport. Please check your reference.
  13. In the sentence“Obesity is a complex and chronic disease that can potentially affect the … and social well-being,” the use of “their” is unclear, as “obesity” is a singular noun. Please revise to ensure grammatical consistency.
  14. The phrase“epidemic disease” is not commonly used in WHO terminology.
  15. “Above 30 kg/m²”should be revised to “≥30 kg/m²” for clinical accuracy and consistency with standard medical guidelines.
  16. Verb tenses are inconsistent in the manuscript. For example, in the sentence“obesity was defined by a body mass index … in contrast to overweight, which is defined by…”, the past tense “was defined” and the present tense “is defined” are mixed. Please unify the tense.
  17. The word“neoplastic” should be used as part of a noun phrase. Consider revising the sentence “There is enough evidence highlighting…” to something like “There is sufficient evidence highlighting the role of obesity in neoplastic diseases…”
  18. The phrase“an increased inflammation” is incorrect, as “inflammation” is uncountable. It should be revised to “increased inflammation.”
  19. In the sentence“Thereby, Chang and cols. described…”, the phrase “drop-in” should be corrected to “drop in.”
  20. The sentence“Excessive insulin secretion favors insulin to interfere with the selective activity of podocytes...” is awkward and unclear. Consider revising for clarity and precision.
Comments on the Quality of English Language

The English should be improved for publication.

Author Response

We would like to express our gratitude to the reviewers and the editorial board for their efforts in reviewing our manuscript entitled “Obesity-Related Kidney Disease: A Growing Threat to Renal Health.” We appreciate the reviewers’ comments, which have undoubtedly enhanced the current version of the manuscript. Below is a detailed description of the changes made to the manuscript.

Reviewer: 2

1. The statement “Adiponectin is primarily found in...” appears to be a misunderstanding. Adiponectin is secreted by adipocytes and circulates in the bloodstream; it exerts its effects via receptors in tissues such as the arterial endothelium, kidney smooth muscle cells, and capillary endothelium.

We greatly appreciate the reviewers' observation. We have revised the phrase to read as follows: “Adiponectin typically facilitates the oxidation of fatty acids and plays a crucial role in the regulation of glucose metabolism. This adipokine is primarily secreted by adipose tissue and is present in the bloodstream. Adiponectin levels are directly associated with renal function. Its physiological effects are mediated through receptors located in various tissues, including the arterial endothelium, smooth muscle cells of the kidney, and capillary endothelium

2. In the sentence “Insulin resistance is directly related to the fat mass of individuals and the secretory activity of adipokines”, does “fat mass” specifically refer to visceral fat mass? Please clarify.

The comment made by the reviewer has enhanced the clarity of the text. We have addressed visceral fat mass in relation to insulin resistance.

3. The sentence “…facilitating the filtration of albumin and promoting oxidative stress through NADPH oxidase and AKT/mTOR intracellular pathway or glucose transporter 4…” combines multiple mechanisms too tightly. Please briefly specify each mechanism. Additionally, please provide a reference supporting the direct involvement of GLUT4 in albuminuria or oxidative stress in the kidney.

Thank you very much for the comment made by the reviewer. We have described the mechanisms separately for better understanding with the following paragraph:

“Podocytes express glucose transporters (GLUT1, GLUT2, GLUT3, GLUT4, and GLUT8) and components of the insulin signaling cascade, such as IRS and the insulin receptor. Podocytes are capable of glucose uptake in response to insulin stimulation through GLUT4, as GLUT4 is the glucose transporter most sensitive to such stimuli. Under normal conditions, insulin induces the rapid translocation of GLUT4 to the plasma membrane, promoting cytoskeletal remodeling and contraction, thereby facilitating the physiological response to glomerular pressure. Renal damage is manifested by the inability to phosphorylate AKT, which prevents the translocation of GLUT4 to the plasma membrane following insulin stimulation and, consequently, decreases glucose uptake. The deficiency of GLUT4 also suppresses the mammalian target of rapamycin (mTOR) pathway, resulting in a deficit in nutrient sensing”.

4. A reference is needed for the statement: “DM has been rising over the past decade, with an estimated 643 million people diagnosed in 2023.”

We appreciate the comment. We have added the reference to the sentence (PMCID: PMC10958528.

5. For the statement “The pancreas adjusts to insulin resistance by initially increasing the number of beta-cells,” please provide a reference. It should be confirmed whether the typical compensatory mechanism involves increased insulin secretion rather than an increase in beta-cell number.

Thank you very much for the correction made by the reviewer. We have modified the explanation in the new manuscript.

6. In the sentence describing the FLOW trial,“The FLOW trial... demonstrated that subcutaneous semaglutide was associated with a risk reduction in MACE in patients with overweight or obesity and established cardiovascular disease without a history of diabetes”, please double-check the trial population. It appears this may be a confusion with the SELECT trial.

We are very grateful for the reviewer's comment. We have reviewed the study population and have modified the sentence as follows: “The FLOW trial (Effect of semaglutide versus placebo on the progression of renal impairment in people with type 2 diabetes and chronic kidney disease) demonstrated that subcutaneous semaglutide was associated with a risk reduction in major adverse renal events (MARE) and death from cardiovascular causes in patients T2DM and CKD”.

7. “CKM syndrome” is a relatively new concept and is not yet a universally established clinical diagnosis. Please confirm whether SGLT2 inhibitors have been specifically indicated for CKM in the cited references.

We appreciate the comment made by the reviewer; however, SGLT2 inhibitors are the cornerstone of treatment for heart failure with preserved and reduced ejection fraction, regardless of albuminuria. Furthermore, they are the treatment of choice in diabetes and chronic kidney disease with eGFR ≥ 20 ml/min/1.73 m² and albuminuria. Although the drug is not specifically targeted at CKM syndrome, it is part of the standard treatment for the previously mentioned conditions on an individual basis

8. The mechanism of SGLT2 inhibitors does not involve Na/K-ATPase in glucose transport. Please verify and revise according to appropriate references.

We appreciate the comment made by the reviewer. We have modified the sentence in the new version of the manuscript as follows: “The success of this class of drugs is based on an insulin-independent mechanism that involves a competitive interaction between the SGLT2 protein and glucose binding in the renal tubules, followed by the transport of glucose across the basolateral membrane to enter the bloodstream”.

1. For the sentence “Obesity is a complex and chronic disease that can potentially affect the … and social well-being”, "their" doesn't have a clear plural referent since the subject is "obesity", which is singular.

We believe that the comment made by the reviewer has contributed to the fluency of the reading. We have replaced the sentence with: “Obesity is a complex and chronic disease that can potentially affect the quality of life by decreasing physical health as well as mental and social well-being

2. “Epidemic disease” is not the common phrasing used by WHO, please replace it.

We are very grateful for the observation; we have replaced “epidemic disease” with “global health epidemic”.

3. “Above 30 kg/m²” should be “≥30 kg/m²” for clinical accuracy.

We appreciate the reviewer's comment and have made the suggested changes.

4. The tenses are mixed in the manuscript, for example, “was defined” and “is defined” is mixed in the sentence “obesity was defined by a body mass index … in contrast to overweight, which is 48 defined by… ”.

The comment made by the reviewer is important; therefore, we have carefully verified the verb tenses.

5. “Neoplastic” should be part of a noun phrase in the sentence “There is enough evidence highlighting…”.

We have made the suggested changes. We appreciate the reviewer's collaboration.

6. “An increased inflammation” is incorrect → “increased inflammation” (uncountable).

The comment made by the reviewer has supported the fluency of the text. We appreciate the effort made by the reviewer.

7. “Drop-in” should be “drop in” in the sentence “Thereby, Chang and cols. Described…”

The observation made by the reviewer is significant, we appreciate it. To avoid confusion, we have modified the phrase to: “a decline in the glomerular filtration rate”.

8. The sentence “Excessive insulin secretion favors insulin to interfere with the selective activity of podocytes...” is awkward

We appreciate the correction made by the reviewer; we have changed the phrase to “Excessive insulin secretion interferes with podocytes and cytoskeleton activities”.

9. Pancreas typically compensates by increasing insulin secretion, rather than the number of beta cells.

Thank you very much for the correction made by the reviewer. We have modified the explanation in the new manuscript.

10. For “The FLOW trial... demonstrated that subcutaneous semaglutide was associated with a risk reduction in MACE in patients with overweight or obesity and established cardiovascular disease without a history of diabetes”, please check patients involved in FLOW trial and whether you confused it with the SELECT trial.

We are very grateful for the reviewer's comment. We have reviewed the study population and have modified the sentence as follows: “The FLOW trial (Effect of semaglutide versus placebo on the progression of renal impairment in people with type 2 diabetes and chronic kidney disease) demonstrated that subcutaneous semaglutide was associated with a risk reduction in major adverse renal events (MARE) and death from cardiovascular causes in patients T2DM and CKD”.

11. “CKM syndrome” is a relatively newly defined construct, not a universally established clinical diagnosis. Has SGLT2i been specified for using in CKM in your reference?

We appreciate the comment made by the reviewer; however, SGLT2 inhibitors are the cornerstone of treatment for heart failure with preserved and reduced ejection fraction, regardless of albuminuria. Furthermore, they are the treatment of choice in diabetes and chronic kidney disease with eGFR ≥ 20 ml/min/1.73 m² and albuminuria.

12. The mechanism of SGLT2 inhibitors is independent of Na/K-ATPase involvement in glucose transport. Please check your reference.

We appreciate the comment made by the reviewer. We have modified the sentence in the new version of the manuscript as follows: “The success of this class of drugs is based on an insulin-independent mechanism that involves a competitive interaction between the SGLT2 protein and glucose binding in the renal tubules, followed by the transport of glucose across the basolateral membrane to enter the bloodstream”.

13. In the sentence “Obesity is a complex and chronic disease that can potentially affect the … and social well-being,” the use of “their” is unclear, as “obesity” is a singular noun. Please revise to ensure grammatical consistency.

We believe that the comment made by the reviewer has contributed to the fluency of the reading. We have replaced the sentence with: “Obesity is a complex and chronic disease that can potentially affect the quality of life by decreasing physical health as well as mental and social well-being

14. The phrase “epidemic disease” is not commonly used in WHO terminology.

We are very grateful for the observation; we have replaced “epidemic disease” with “global health epidemic”.

15. “Above 30 kg/m²” should be revised to “≥30 kg/m²” for clinical accuracy and consistency with standard medical guidelines.

We appreciate the reviewer's comment and have made the suggested changes.

16. Verb tenses are inconsistent in the manuscript. For example, in the sentence“obesity was defined by a body mass index … in contrast to overweight, which is defined by…”, the past tense “was defined” and the present tense “is defined” are mixed. Please unify the tense.

The comment made by the reviewer is important; therefore, we have carefully verified the verb tenses.

17. The word “neoplastic” should be used as part of a noun phrase. Consider revising the sentence “There is enough evidence highlighting…” to something like “There is sufficient evidence highlighting the role of obesity in neoplastic diseases…”

We have made the suggested changes. We appreciate the reviewer's collaboration.

18. The phrase “an increased inflammation” is incorrect, as “inflammation” is uncountable. It should be revised to “increased inflammation.”

The comment made by the reviewer has supported the fluency of the text. We appreciate the effort made by the reviewer.

19. In the sentence “Thereby, Chang and cols. described…”, the phrase “drop-in” should be corrected to “drop in.”

The observation made by the reviewer is significant, we appreciate it. To avoid confusion, we have modified the phrase to: “a decline in the glomerular filtration rate”.

20. The sentence“Excessive insulin secretion favors insulin to interfere with the selective activity of podocytes...” is awkward and unclear. Consider revising for clarity and precision.

We appreciate the correction made by the reviewer; we have changed the phrase to “Excessive insulin secretion interferes with podocytes and cytoskeleton activities”.

Reviewer 3 Report

Comments and Suggestions for Authors

This is an interesting review of the interplays among obesity, diabetes, cardiovascular disease, and chronic kidney disease. It also explores the known and potential underlying mechanisms.

The review is well-written and well-organized by categories in most of the sections. However, the authors might also want to emphasize the relationship between obesity and DM.

The authors should claim whether generative AI was used to generate data and figures, which could be questioned.

  1. Line 47, please list some important “socio-environmental factors”
  2. Line 50-51, using “combination of “rather than “in contrast, the addition of” might be better.
  3. Line 52-54, need to provide references.
  4. Line 55-60, The statement is from the same reference, but sounds like it comes from different refs. Also need to explain “fat mass disease” and sick fat disease”.
  5. Line 66, Should use Chang et al. The “Author Name and cols” has been used several times.
  6. Line 68, What is the meaning of “drop-in”?
  7. Line 79, need to list the “previously mentioned pathways” to avoid confusion.

Author Response

We would like to express our gratitude to the reviewers and the editorial board for their efforts in reviewing our manuscript entitled “Obesity-Related Kidney Disease: A Growing Threat to Renal Health.” We appreciate the reviewers’ comments, which have undoubtedly enhanced the current version of the manuscript. Below is a detailed description of the changes made to the manuscript.

Reviewer: 3

1. Line 47, please list some important “socio-environmental factors”

Thank you very much to the reviewer for the comment provided. We have described some factors within the paragraph.

2. Line 50-51, using “combination of” rather than “in contrast, the addition of” might be better.

The comment has improved the fluency. We greatly appreciate the feedback provided.

3. Line 52-54, need to provide references.

We appreciate the comment and have included references in the mentioned lines.

4. Line 55-60, The statement is from the same reference, but sounds like it comes from different refs. Also need to explain “fat mass disease” and sick fat disease”.

The comments provided by the reviewer have been extremely helpful in improving the grammar and fluency of the paragraph. We have made the requested changes and clarified the relevant terms.

5. Line 66, Should use Chang et al. The “Author Name and cols” has been used several times.

We appreciate the suggestion provided from the revisor. We already change to Chang et al. in the new version of the manuscript.

6. Line 68, What is the meaning of “drop-in”?

The observation made by the reviewer is significant, we appreciate it. To avoid confusion, we have modified the phrase to: “a decline in the glomerular filtration rate”.

7. Line 79, need to list the “previously mentioned pathways” to avoid confusion.

We appreciate the revisor comment and have revised the sentence to avoid any confusion within the manuscript.

Round 2

Reviewer 1 Report

Comments and Suggestions for Authors

The manuscript has been revised adequately. Overall well written.

Some minor errors to be corrected - e.g. line 187 "Effect of obesity (on) kidney disease"